# A climate suitability index for species distribution modelling applied to terrestrial arthropods in the Mediterranean Region

James M. Ciarlo`[1,2], Monique Borg Inguanez[3], Erika Coppola[2], Aaron Micallef[4,5], David Mifsud[1]

[1]Institute of Earth Systems, University of Malta, Msida, Malta
[2]Abdus Salam International Centre for Theoretical Physics, Trieste, Italy
[3]Department of Statistics and Operations Research, University of Malta, Msida, Malta
[4]Department of Geosciences, University of Malta, Msida, Malta
[5]Monterey Bay Aquarium Research Institute, Moss Landing, CA, USA

*Correspondence to*: James M. Ciarlo` (james.ciarlo[at]um.edu.mt)

**Abstract.** Climate change poses significant threats to global biodiversity, particularly impacting arthropods due to their sensitivity to shifts in temperature and precipitation, as well as other environmental conditions. These changes impact the suitability of their habitats, alter ecological interactions, and consequently affect the distribution and survival of species. Understanding how climate variability influences the ecological niches of arthropods is crucial for predicting future biodiversity patterns and implementing effective conservation strategies. This study introduces a simple index designed to model species' distribution on the basis of their climatic niche, with a specific focus on terrestrial Mediterranean arthropods. This approach leverages Regional Climate Model data to construct a climatology of a species's preferred habitat, based on historically observed locations. This index offers a straightforward and rapid means to assess the resilience and vulnerability of arthropod populations, and could be applied to future studies aiming to shed light on how climate change could affect the fundamental niches of terrestrial arthropods. The analysis revealed that the method is most reliable for species with observations exceeding 1000 points, and climate datasets of high resolutions (although the latter had a smaller influence on the results). This study offers a proof-of-concept for the proposed index, demonstrating its potential utility in guiding conservation strategies and mitigating the adverse effects of climate change on arthropod habitats.

## 1    Introduction

Arthropods are the largest and most diverse group of animals on Earth. They occupy nearly every ecological niche and are found in almost all terrestrial and aquatic habitats (Gullan and Cranston, 2014; Kotze et al., 2022; Schowalter, 2022). Arthropods play essential roles in maintaining ecosystem health and stability, serving as pollinators, predators, detritivores, and other important roles within their diverse habitats (Chakravarthy et al., 2016; Gullan and Cranston, 2014; Rundel and Gibson, 1996; Schowalter, 2022). Hence, they are present at various levels of the food web, and many are extremely sensitive to changes in their environment, whose effects can quickly propagate up the food web. As a consequence of all these factors, many arthropods can act as indicators of ecosystem integrity (Maleque et al., 2006). The state of these

ecosystems is often sensitive to variations in climate conditions, especially in the Mediterranean basin (Gritti et al., 2006; Santos et al., 2014; Vogel et al., 2021). In recent decades, the diversity of insect pollinators has faced numerous threats due to changes in the environment (Arce et al., 2023; Forister et al., 2021; Raven and Wagner, 2021; Wagner et al., 2021; Zattara and Aizen, 2021), among which climate change emerges as one of several important stressors (Botsch et al., 2024; Outhwaite et al., 2022; Potts et al., 2016; Uhl et al., 2022).

The ecological impacts of the climate crisis vary across the globe (Chen et al., 2021; Cui et al., 2021; Eyring et al., 2016), especially in vulnerable regions such as the Mediterranean basin (Giorgi, 2006; Lionello and Scarascia, 2018; Ranasinghe et al., 2021), and its numerous small islands. According to the Sixth Assessment Report (AR6) by the Intergovernmental Panel for Climate Change (IPCC) droughts in the Mediterranean are already increasing, and the basin is projected to become increasingly arid together with a rise in extreme temperature (Doblas-Reyes et al., 2021; Gutiérrez et al., 2021; Ranasinghe et al., 2021). The impact of these changes on the ecosystem varies according to numerous factors, and the extent to which insects and other arthropods are affected remains uncertain (Arce et al., 2023). This is especially so when changes to a particular group of organisms (such as pollinators) can impact other members of the ecosystem (Mullin et al., 2023).

One approach to study the climate impacts on arthropods and their habitats is to map species distribution with the use of Ecological Niche Modelling (ENM; Fletcher Jr. et al., 2019; Haase et al., 2021; Hiller et al., 2019; Mammola et al., 2021; Mugumaarhahama et al., 2023; Phillips et al., 2004; Sillero et al., 2023; Tesfamariam et al., 2022). This approach offers the possibility of predicting potential shifts in species distributions under future climate scenarios, thereby providing valuable insights into the resilience and vulnerability of arthropod populations and their ecosystems. However, ENM can be especially challenging, when considering accurate presence-absence data, and additional non-climate factors that determine the distribution of a particular species (e.g. presence of predators, specific plants, competitors, and land-use). While access to climate data has become increasingly available (Mammola et al., 2021), this also has its limitations, as very high-resolution data (e.g. CHELSA with ≈1 km spatial resolution; Karger et al., 2017) is preferred. These datasets are not abundant, their temporal coverage is limited, as is their range of variables.

Some ecological studies (Adão et al., 2023; Fink and Scheidegger, 2018; Khan et al., 2020; Mauri et al., 2022), like those assessed in the AR6, have leveraged the extensive collection of Regional Climate Models (RCMs) from the Coordinated Regional Climate Downscaling Experiment (CORDEX; Coppola et al., 2021; Giorgi, 2014; Giorgi et al., 2009, 2022; Gutowski Jr. et al., 2016; Teichmann et al., 2021), driven by the Coupled Model Intercomparison Project (CMIP; Eyring et al., 2016; Meehl et al., 1997, 2000, 2007; Taylor et al., 2012). Models from these datasets (accessible on the Earth System Grid Federation), such as the EURO-CORDEX (Coppola et al., 2021a; Jacob et al., 2014, 2020) at ≈12.5 km spatial resolution, have undergone thorough validation and offer a wide range of climate variables. RCMs offer a higher resolution compared to global datasets, and excel in representing the climate of complex regions, such as the Mediterranean (Ciarlo` et

al., 2021; Giorgi et al., 2022b; Reale et al., 2022; Somot et al., 2018). Moreover, with recent advances in Convection Permitting (CP) simulations, which offer resolutions of approximately 3 km, the development of kilometre-scale RCM ensembles with diverse variables is within reach (Ban et al., 2021; Coppola et al., 2020; Pichelli et al., 2021). Although most arthropods are relatively small-in-size and tend to occupy regions of specific microclimates (Clark and Johnson, 2024; Høye and Forchhammer, 2008), RCMs' ability to accurately depict climate variations of complex landscapes provides a good understanding of how such organisms may respond to climate change.

This study utilises RCM data to evaluate the influence of climate parameters on terrestrial arthropod habitats, introducing a novel, simplified index for this purpose. The methodology hinges on analysing the climatology of sites where specific species have been documented, and, by integrating RCM data from various time periods and spatial resolutions, it could offer insights into potential shifts in the fundamental niches of these species, or stress exerted by a changing climate. The fundamental niche, which describes an environment that an organism could survive in (but may not necessarily be present in), is represented by climatic (abiotic) parameters within this study. This differs from the realised niche, which is more restricted since it describes where the organism is actually present due to the inclusion of biotic factors (Putman and Wratten, 1984), which are not represented in the RCM data. The findings detailed herein provide a proof-of-concept for this index and demonstrate its applicability in modelling arthropod species distribution on the basis of their climatic niches.

## 2    Data & Methods

This study introduces a new simple metric designed to quantify the climate's influence on the distribution of certain terrestrial arthropods, a tool that can be critical in the future, given the anticipated direct impacts of climate change (IPCC, 2023). This metric is based on the assumption that a living organism observed at a specific location will have favourable climatic conditions for its existence. Hence, a collection of locations where the organism was observed can describe the range of climate parameters of its fundamental niche.

For a potential species of interest (PSI; e.g., *Spilostethus pandurus*) $s$, with $n_s$ sampling/observation locations, and a selection of climate indices (see Section 2.2), the value of an index at a sample location can be expressed as $x_{sij}$ where $i$ represents a specific climate index (examples of such indices include annual mean of near-surface air temperature [*tasmean*], or annual sum of precipitation [*prsum*]) such that $i=1,...p$; $p$ denotes the number of indices considered, and $j$ represents a specific location such that $j=1,...n_s$. The corresponding mean for the $i^{th}$ index of the population of $s$ can be expressed as $\mu_{si} = \frac{1}{n_s}\sum_{j=1}^{n_s} x_{sij}$. The most appropriate conditions for $s$ would occur when $x_{sij}$ approaches the value of $\mu_{si}$ (difference at, or close to, 0), hence we can define the preferred climate conditions, $C_{si}$, to be maximal (i.e. 1). As $x_{sij}$ deviates from $\mu_{si}$, the climate index identifies less favourable conditions, until it exceeds the limit, $L_{si}$. Thus, $C_{sij}$ can be expressed as Equation (1) below.

$$
(1) \qquad C_{sij} = \begin{cases} 1, & \text{if} |x_{sij} - \mu_{si}| = 0 \\ 1 - \left|\dfrac{d_{sij}}{L_{si}}\right|, & \text{if} |x_{sij} - \mu_{si}| = d_{sij}\sigma_{si} \\ 0, & \text{if} |x_{sij} - \mu_{si}| = L_{si}\sigma_{si} \end{cases}
$$

Using Equation (1), where $\sigma_{si}$ is the standard deviation of the $i^{\text{th}}$ index for the population $s$, and $d_{sij}$ is the standardised distance to the mean, $(x_{sij} - \mu_{si})/\sigma_{si}$, $C_{sij}$ can be reduced to Equation (2).

$$
(2) \qquad C_{sij} = 1 - \left|\frac{x_{sij} - \mu_{si}}{\sigma_{si}}\right| \frac{1}{L_{si}}
$$

The limit, $L_{si}$, is expressed as Equations (3) and (4), which describe the largest deviation from the maximum or minimum of $d_{sij}$.

$$
(3) \qquad L_{si} = max(d_{si,max}, d_{si,min})
$$

$$
(4) \qquad d_{si,max} = \left|\frac{x_{si,max} - \mu_{si}}{\sigma_{si}}\right|, \quad d_{si,min} = \left|\frac{x_{si,min} - \mu_{si}}{\sigma_{si}}\right|
$$

The different quantities of $C_{sij}$ are combined into the **Eco-Climate Index** for species $s$ at location $j$, $EI_{sj}$, which describes the
105 climatological component of a species's ecological niche, as shown in Equation (5). The value of $EI_{sj}$ is expressed relative to the maximum of all combined $C_{si}$ at each location $j$ (only for existing observation) to normalise the index. This produces a quantity that ranges between 0 and 1, where 0 describes climate conditions beyond the accepted limit for $s$, and 1 describes the apparent ideal climate conditions for $s$ according to its sampling locations. It is important to note that a value of 1 does not imply the presence of $s$ as non-climatological factors (e.g., human influence, presence of competitors, availability of
110 food) are not included in this metric. Since $EI_{sj}$ refers to the Eco-Climate Index of species $s$ at location $j$, when referring to spatial maps this becomes $EI_s$

$$
(5) \qquad EI_{sj} = \frac{C_{s1j} \times ... \times C_{spj}}{max(C_{s1j} \times ... \times C_{spj})}
$$

## 2.1 Biodiversity Data

In order to evaluate the Eco-Climate index introduced in Equation (5), an analysis was focused on the broader European region. This permitted the use of RCM data from the EURO-CORDEX ensemble, as well as a new ≈3 km CP simulation of the western and central Mediterranean (both described in Section 2.2). The analysis focused on terrestrial species occurring in the European and Mediterranean regions, and the data consisted of research-grade observations from the iNaturalist (iNaturalist community, 2023) database.

For the purposes of this study, eight arthropods (listed in Table 1) were selected as PSIs, where each play important roles in the ecosystem, such as pollinators, predators, herbivores, and detritivores. Each species was selected to cover a variety of observed Mediterranean arthropods from different orders with varied number of occurrences and ecological roles. One

species, *Brachytrupes megacephalus*, was also chosen due to its status as a vulnerable (Buzzetti et al., 2016). The results of this analysis would depend greatly on $n_s$ (some, such as *Brachytrupes megacephalus*, have a very small number of observations). Having small values for $n_s$ can produce less reliable results when determining preferred habitats for PSIs. For this reason, this study also provides a comparative assessment of how variation in $n_s$ influences the product of this metric. Techniques that artificially inflate the sample size, such as bootstrapping, were found to have minimal effect on results and hence were not included to avoid adding unnecessary complexities to the metric.

Table 1: Scientific names and order of the selected PSIs, together with the corresponding number of research-grade observations accessed from the iNaturalist (iNaturalist community, 2023) **database (number of occurrences, $n_s$), and additional rationale for selection.**

| s | Scientific name (authority) | Order | $n_s$ | Additional Rationale |
|---|---|---|---|---|
| 1 | *Ameles decolor* (Charpentier, 1825) | Mantodea | 778 | predator |
| 2 | *Argiope lobata* (Pallas, 1772) | Araneae | 3062 | Arachnida; predator |
| 3 | *Brachytrupes megacephalus* (Lefèvre, 1827) | Orthoptera | 26 | vulnerable; omnivore |
| 4 | *Polyommatus celina* (Austaut, 1879) | Lepidoptera | 631 | pollinator |
| 5 | *Scarabaeus variolosus* (Fabricius, 1787) | Coleoptera | 143 | detritivore |
| 6 | *Selysiothemis nigra* (Vander Linden, 1825) | Odonata | 529 | predator |
| 7 | *Spilostethus pandurus* (Scopoli, 1763) | Hemiptera | 5037 | mainly herbivore |
| 8 | *Xylocopa violacea* (Linnaeus, 1758) | Hymenoptera | 5420 | pollinator |

## 2.2 Climate Data

The purpose of $EI_s$ is to evaluate the climate influence on the fundamental niche of a particular organism, and hence the choice of climate parameters is essential. Several climate indices (Coppola et al., 2021a; Giorgi et al., 2011, 2018; Schwingshackl et al., 2021; Sylla et al., 2018) of varying complexity were considered (see Supplementary Information), but ultimately eight were selected (described in Table 2). The first variables considered to assess the environmental conditions preferred by a given species were temperature (due to its importance to an organism's metabolism) and precipitation (due to the importance of a water source). Given the importance of these variables and their variability throughout the year, the mean conditions together with upper and lower extreme conditions were also deemed important. Thus, indices were selected that represent these conditions for both temperature and precipitation. Given the size of arthropods, average windspeed was also included. Finally, as organisms are known to have a preference to specific altitudes, elevation was also included. Beyond the proof-of-concept these criteria can be used to list starting indices but should not be used as strict rules to be satisfied. It is important to note that this study adheres to these eight indices for the purposes of a homogeneous analysis; however, this metric may be used with any number of climate indices.

**Table 2: The eight climate indices used in this study to describe the climatological component of an ecological niche.**

| $i$ | Short Name | Long Name | Units |
|---|---|---|---|
| 1 | tasmean | Annual mean of near-surface air temperature | °C |
| 2 | cwfi | Cold-wave Frequency Index: Annual mean of 6+ consecutive days below 5-day 10th percentile temperature | days |
| 3 | hwfi | Heat-wave Frequency Index: Annual mean of 6+ consecutive days above 5-day 90th percentile temperature | days |
| 4 | prsum | Sum of Annual precipitation | mm |
| 5 | cdd | Annual mean of maximum consecutive dry days | days |
| 6 | rx1day | Maximum 1-day precipitation in time period | mm/day |
| 7 | windmean | Annual mean of near-surface wind speed | m/s |
| 8 | orog | Surface Altitude | m |

The selection of climate indices was also based in-part on the parameters available from the climate observation dataset used for the analysis. The observations are the 31-year (1980-2010) daily variables of E-OBS v25e at 10° horizontal resolution (Cornes et al., 2018; Haylock et al., 2008), hereafter referred to as E-OBS.

The analysis was extended beyond the observation dataset to the 12 km EURO-CORDEX simulations (available on the Earth System Grid Federation), to showcase the application of this metric to climate models. An ensemble was constructed from simulations driven by the European Centre for Medium-Range Weather Forecasts ERA-Interim reanalysis (ECMWF-ERAINT; Dee et al., 2011). The EURO-CORDEX data is extensive and has been evaluated in numerous studies (Casanueva et al., 2016; Fantini et al., 2018; Kotlarski et al., 2014; Prein et al., 2016; Vautard et al., 2013), and the recent CMIP5-driven
simulation members, which were used in this paper, have been extensively evaluated for temperature, precipitation, winds, and other variables (Ciarlo` et al., 2021; Molina et al., 2023; Sørland et al., 2021; Vautard et al., 2021). This ensemble was constructed only from simulations which provided the parameters necessary to construct the indices described in Table 2 (for the 1980-2010 time period), and availability through ESGF nodes at the time data collection began. The 6 RCMs that satisfied these criteria were selected for this ensemble, which is hereafter referred to as Ens6 (detailed in Table 3). All
ensemble members are at the same spatial resolution, however 2 members (CNRM-ALADIN63, and ICTP-RegCM4-6) required interpolation from their native grid to a common grid using a nearest-neighbour approach. To minimize errors, the indices listed in Table 2 were calculated individually before any interpolation. Furthermore, the final Ens6 product was obtained with the use of an ensemble mean of the indices associated with each member.

**Table 3: A description (including the model, reference, and institute that ran the simulation) of the RCM simulations driven by ECMWF-ERAINT making up Ens6.**

| Institute | RCM | Reference |
|---|---|---|
| CLMcom-ETH | COSMO-crCLIM-v1-1 | Sørland *et al.*, 2021 |
| CNRM | ALADIN63 | Nabat *et al.*, 2020 |
| GERICS | REMO2015 | Jacob, 2001; Jacob *et al.*, 2012 |
| ICTP | RegCM4-6 | Giorgi *et al.*, 2014 |
| KNMI | RACMO22E | van Meijgaard *et al.*, 2012 |
| SMHI | RCA4 | Kupiainen *et al.*, 2011 |

The metric was also applied to a new ≈3 km resolution CP simulation of the western and central Mediterranean (hereafter referred to as WMD03). This new simulation was run using the fifth-generation regional climate modelling system, RegCM5 (Coppola et al., 2024; Giorgi et al., 2023), and driven by the fifth-generation ECMWF reanalysis for the global climate and weather (ECMWF-ERA5; Hersbach et al., 2020), and a parent ≈12 km EURO-CORDEX domain. Both the parent and CP simulations have been run with the non-hydrostatic Moloch core (Davolio et al., 2020; Malguzzi et al., 2006), and physics configuration as presented in Coppola et al., (2024) with the following differences: NoTo microphysics (Nogherotto et al., 2016), Xu & Randall (1996) cloud fraction, and Biosphere-Atmosphere Transfer Scheme land surface module (Dickinson et al., 1993)(Dickinson et al., 1993). The final CP simulation covers a 10-year period (1995-2004), which was included in the analysis to evaluate the performance of the metric for a very-high resolution climate dataset.

The eight climate indices obtained from the E-OBS dataset were computed for these modelled datasets and evaluated using the E-OBS-derived indices as a reference dataset. The comparison was performed with the use of a standard percentage bias as described in Equation (6).

$$(6) \qquad \frac{(Model - Reference)}{Reference} \times 100$$

This data is brought together in the analysis with two time periods: 31-years (1980-2010) for the assessment of the observation and RCM data; and 10-years (1995-2004) for the inclusion and comparison of the CP data. The latter 10-year time period is also applied to the observation and RCM data, and while the individual 10-year assessment is shown in the Supplementary Information, the direct comparison of all datasets was performed on this common time period.

## 3 Data Analysis

### 3.1 Climate Indices

The EcoClimate Index needs to be constructed using climate indices that represent the environmental conditions of an arthropod's habitat. Therefore, the climate indices listed in Table 2 should represent the climatological component of an ecological niche in order to be used in the evaluation of this metric. In order to avoid cases of double sampling, correlations between prospective indices (described in Table 2 and S1) were analysed and only those with a correlation lower than 0.5 were selected for this analysis (the matrix of scatter plots and a summary of the correlation coefficients is presented in Figure S1 and Table S2). This study, serving as a proof-of-concept for this metric, was designed for a homogeneous inter-species assessment, and hence this correlation limit was considered an acceptable constraint. However, for targeted in-depth analysis of individual species using the metric, it is advisable to construct the index from environmental parameters that are as independent as possible from those of other species. Ideally, these indices should exhibit even lower correlations than the set threshold to ensure greater precision.

The metric of $EI_s$ (described in Section 2) is also computed based on two RCM datasets, a 12 km Europe ensemble of 6 simulations (Ens6), and a 3 km CP simulation of the western and central Mediterranean (WMD03). The percentage biases of each index (shown in Figure 1 and Figure 2, as well as Figures S27 to S32 for each ensemble member) reveal limitations of the modelled datasets. The most prominent bias for Ens6 is a prominent wet bias for rx1day mostly in Spain, South Italy, and the East of the domain. This is consistent throughout all models but to a lesser extent for RegCM and RACMO. A wetter (prsum) and colder (tasmean) bias is noticeable in mountainous regions, consistent with observational challenges associated with these areas, as noted in other studies (Adam and Lettenmaier, 2003; Ciarlo` et al., 2021). All members of Ens6 also show a positive bias over Central Europe for cwfi and especially hwfi, as well as a windy bias throughout the domain. The biases for the WMD03 dataset are more pronounced than the Ens6. This could be partially attributed to the model configuration and/or increased resolution revealing similar issues to mountainous regions related to station density within the reference dataset (a more detailed assessment of the CP dataset is upcoming in a separate study). While these biases will certainly influence the values of $EI_s$, this might not be prominent (see below) since the $EI_s$ is based on the statistical descriptors ($\mu_{si}$, $\sigma_{si}$, $L_{si}$) of the climate indices calculated within each location. This is noticeable in the actual values obtained for all statistical descriptors (see Supplementary Information Tables S3 to S7) compare well within datasets, revealing that the datasets are still comparable for the purposes of this study.

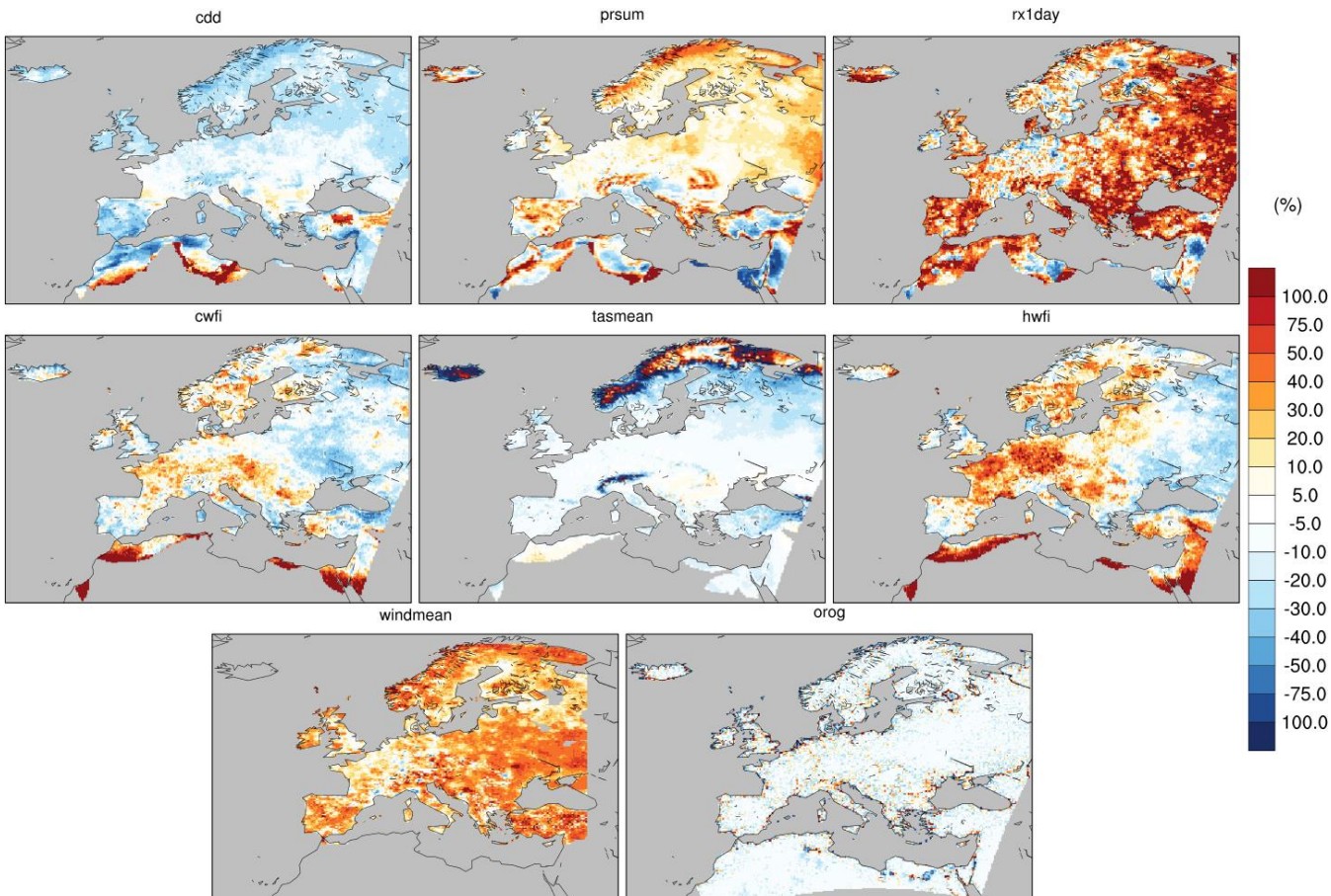

**Figure 1: Percentage bias for climate indices from the 31-year Ens6 (see Table 3) compared to E-OBS dataset (1980-2010).**

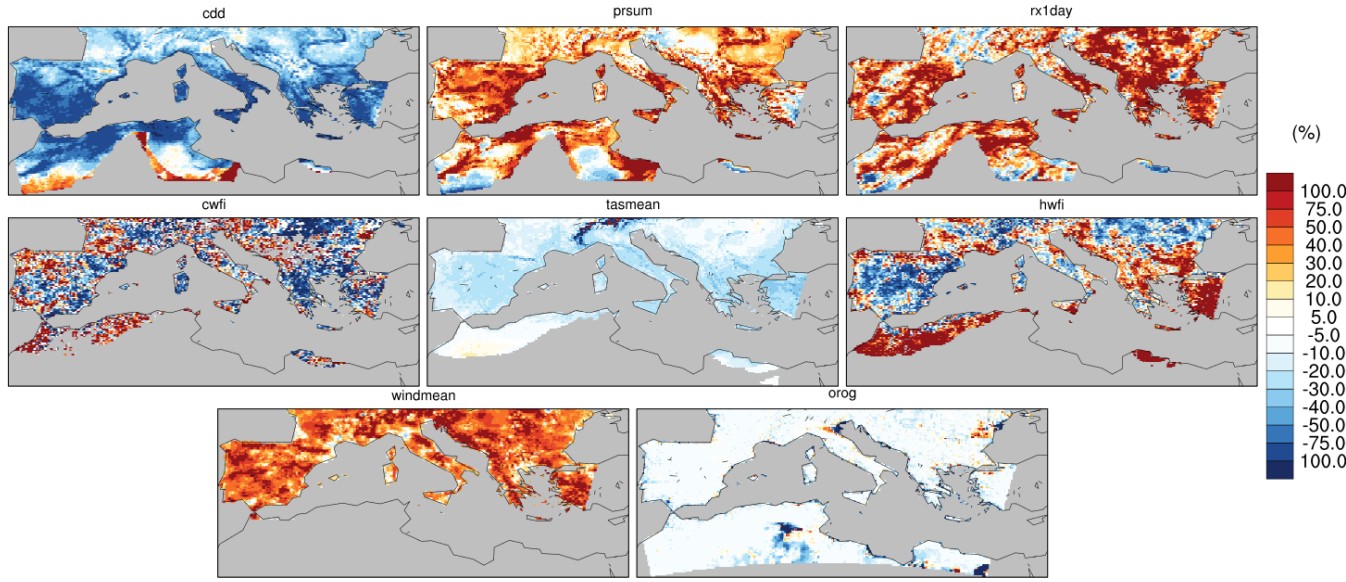

**Figure 2: Percentage bias for climate indices from the 10-year WMD03 (driven by ERA5) compared to E-OBS dataset (1995-2004).**

## 3.2 Eco-Climate Index analysis

To demonstrate the application of the Eco-Climate Index, described in Section 2, the E-OBS climate dataset was utilised first. *Spilostethus pandurus* was selected as the first case study, with over 5000 iNaturalist observations. This approach provides a detailed illustration of the index's capabilities, with results summarised in Figure 3, reflecting the index's

performance using extensive empirical data. The spatial maps shown in Figure 3a-h illustrate the eight climate indices (expressed in terms of the preferred climate conditions, $C_{si}$) as separate components of the fundamental niche, each ranging from 0 to 1, which represent the worst and best state of the index respectively. For any PSI, each component can be analysed in detail, if necessary, for example, Figure 3a shows a broad area with climate conditions at or approaching 1. In this case, given that the lower limit of cdd is 0 days, these minimal drought conditions likely do not pose any additional stress to a PSI.

The observation locations of *Spilostethus pandurus* $n_s$ (Figure 3i) is different from that given in Table 1. This is because some of the original 5037 points correspond to grid-cells not included in the E-OBS dataset (represented as 'miss.' in Figure 3j) and thus, in this case, $n_s$ is reduced to 3644. When the spatial maps of Figure 3a-h are combined, the Eco-Climate indices of the species, $EI_s$ (Figure 3k) is obtained. This spatial map thus describes the fundamental niche for *Spilostethus pandurus*

according to the observed locations of iNaturalist and the climate conditions of E-OBS.

It should be noted that the value of $EI_s$ extends to areas where no observations can be found. This does not imply that these are previously unknown habitats of *Spilostethus pandurus*, but rather that this represents the fundamental niche for the species, and hence favourable climate conditions for the organism. The interaction with other species (host plants, predator-prey relationships, human presence) is not included in this metric and therefore it cannot describe the realised niche.

While the spatial distribution for $EI_s$ (Figure 3k) is appreciably similar to the spatial distribution of the points of observation (Figure 3j), not all points result in a high $EI_s$. Figure 3k also includes points of observed locations where the corresponding $EI_s$ value is less than 0.1, i.e. regions with the least likely chance of observation. This is quantified with the term $p_{0.1}$, which describes the percentage of valid points within this threshold, and is thus used as a measure of the metric's "effectiveness" in this study.

Although the value of $EI_s$ is a unitless quantity, it is constructed from statistical descriptors of climate parameters (as described above). The method allows for the determination of the ideal conditions of each climate index (in their respective units) associated with a particular species. All actual values of statistical descriptors associated with every dataset and time period analysed within this study are defined in Tables S3 to S7 of the Supplementary Information. It's important to note that for the purposes of this study, these values represent a confirmation that no abnormal conditions are being considered. However, these quantities should not be interpreted as the actual parameters preferred by each species, since as described earlier, the climate indices were not selected to represent the precise preferences of each species but to allow for a homogenous assessment of the $EI_s$ metric.

The distributions shown in Figure 3 could be applied within the context of a different study to extrapolate on expected impacts due to climate change. For instance, with the expected aridification of the Mediterranean (IPCC, 2023), one would expect that some impacted regions would become less hospitable to a PSI based on the importance of prsum (e.g., Figure 3b). Another example based on plots of tasmean (e.g., Figure 3e), could show northward migration of the ideal temperature conditions, as southern regions become less hospitable with increasing temperatures.

# Spilostethus pandurus

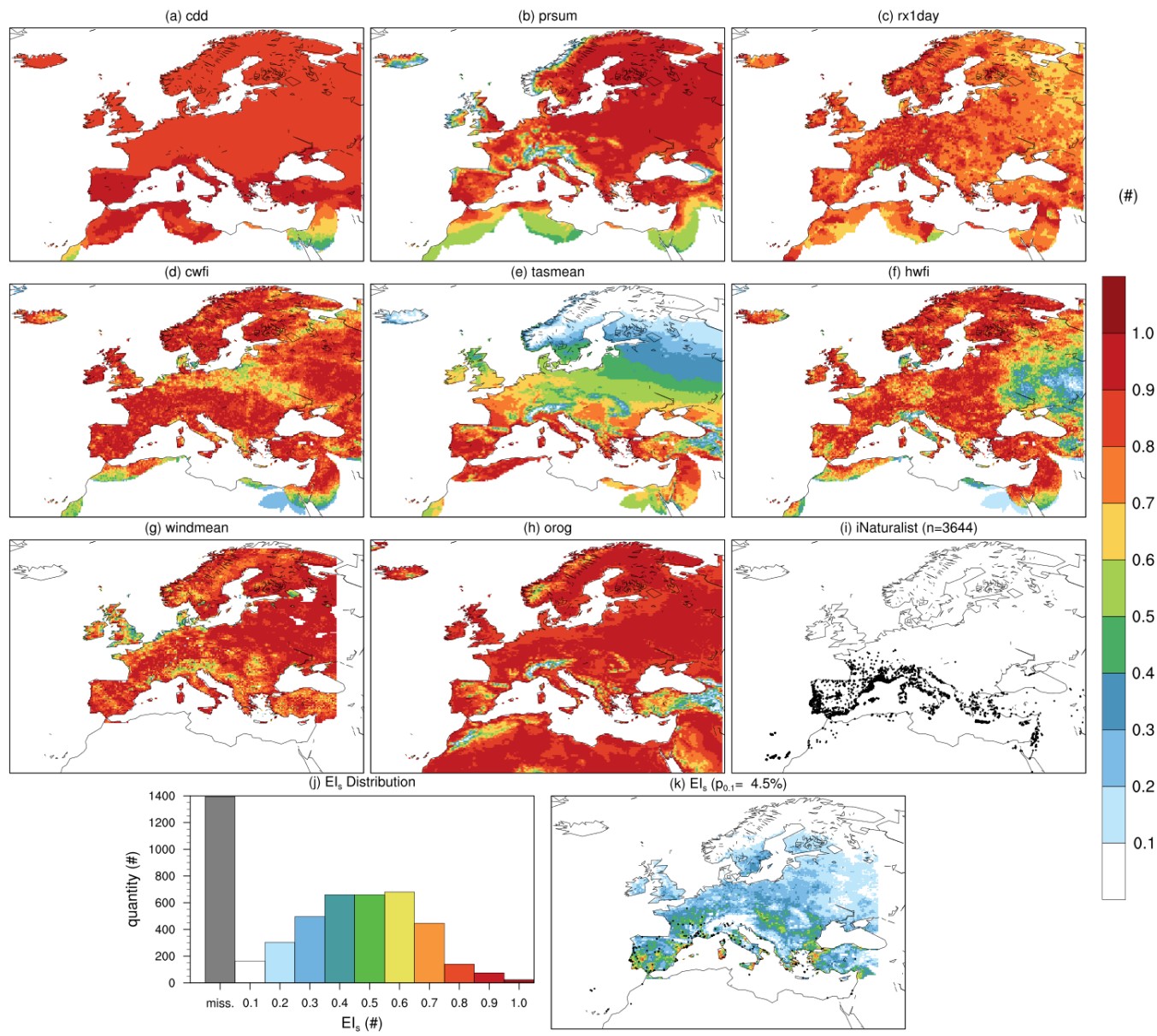

**Figure 3: The EIs product and components for *Spilostethus pandurus* according to the 1980-2010 E-OBS dataset. (a-h) The climate indices expressed in terms of affinity to *Spilostethus pandurus*. (i) The observation points (iNaturalist) and quantity, *n*, applied. (j) The distribution of *EI_s* values, including the number of points from the original dataset that could not be applied ("miss."). (k) The spatial distribution of *EI_s*, including the points less than 0.1 (quantified with percentage $p_{0.1}$).**

The number of points, $n_s$, used in the initial assessment of the habitat could also influence $p_{0.1}$. The different PSIs listed in Table 1 provide the opportunity to evaluate the metric for datasets with different $n_s$. A summary of the spatial distributions of

*EI$_s$* for all eight PSIs is shown in Figure 4 and the corresponding images analogous to Figure 3 are shown in Figures S2-S8. When investigating the lowest value for $n_s$, too few points from the *Brachytrupes megacephalus* dataset could be used with the E-OBS dataset, resulting in a constant *EI$_s$* field of 0 (these results were maintained for consistency throughout this paper). PSIs with $n_s$ greater than 100 have $p_{0.1}$ values of ≈20%, while those with $n_s$ greater than 1000 have $p_{0.1}$ values between 4-6%. This suggests that with higher $n_s$, the method becomes more effective at reproducing the fundamental niche.

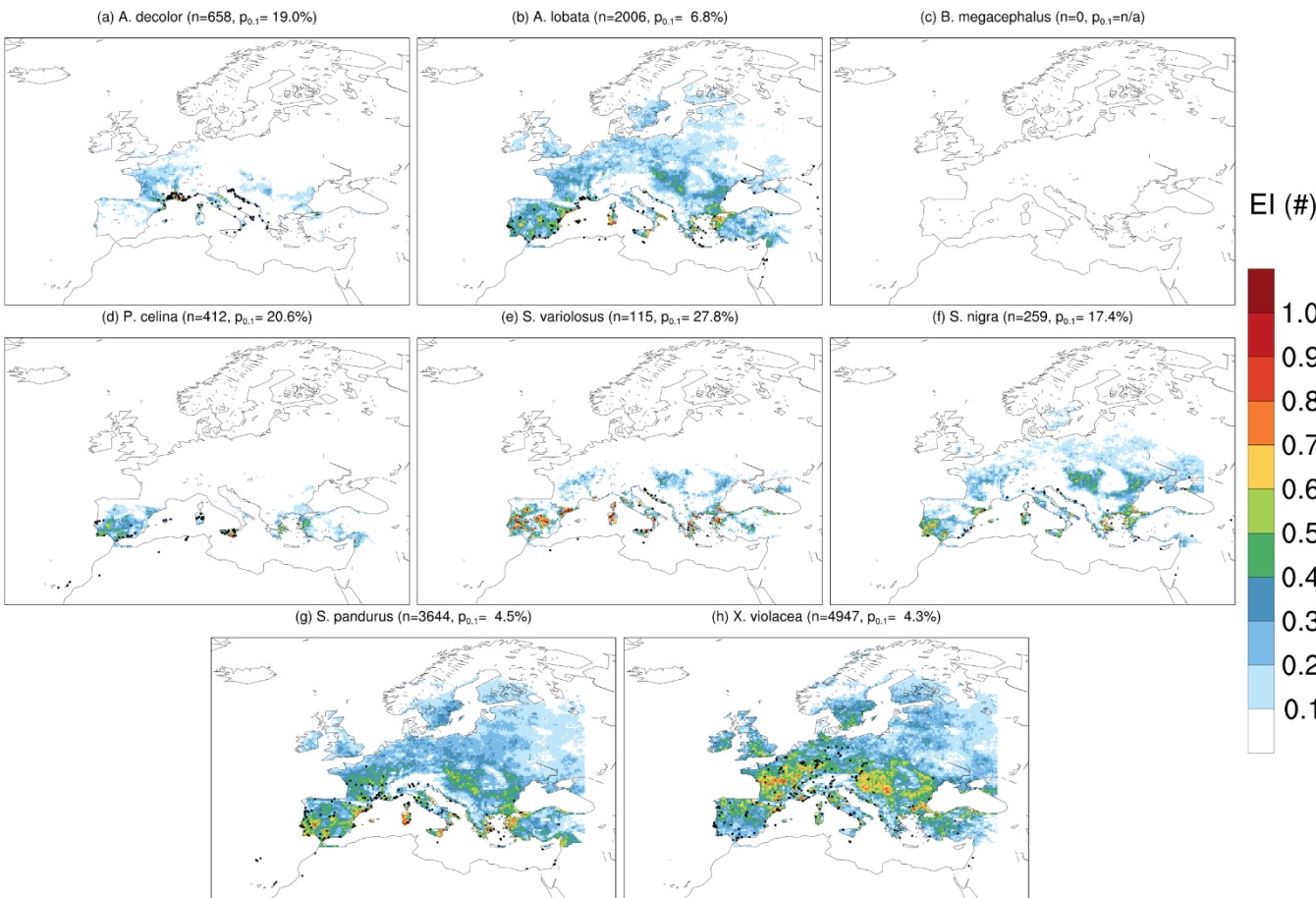

**Figure 4. The spatial distribution of *EI$_s$* for all eight PSIs applied to the 1980-2010 E-OBS dataset. Individual products of each PSI are shown in Figure 3 and Figures S2-S8.**

Considering that several observations correspond to coastal areas or small islands, they cannot be properly represented within the E-OBS dataset as it is limited to the land. This is explored with the use of the Ens6 data (Figure 5) which makes use of all iNaturalist coordinates within the EURO-CORDEX region. The results, while similar, do not reduce the value of $p_{0.1}$, and the small differences obtained may be due to model biases (Figure 1).

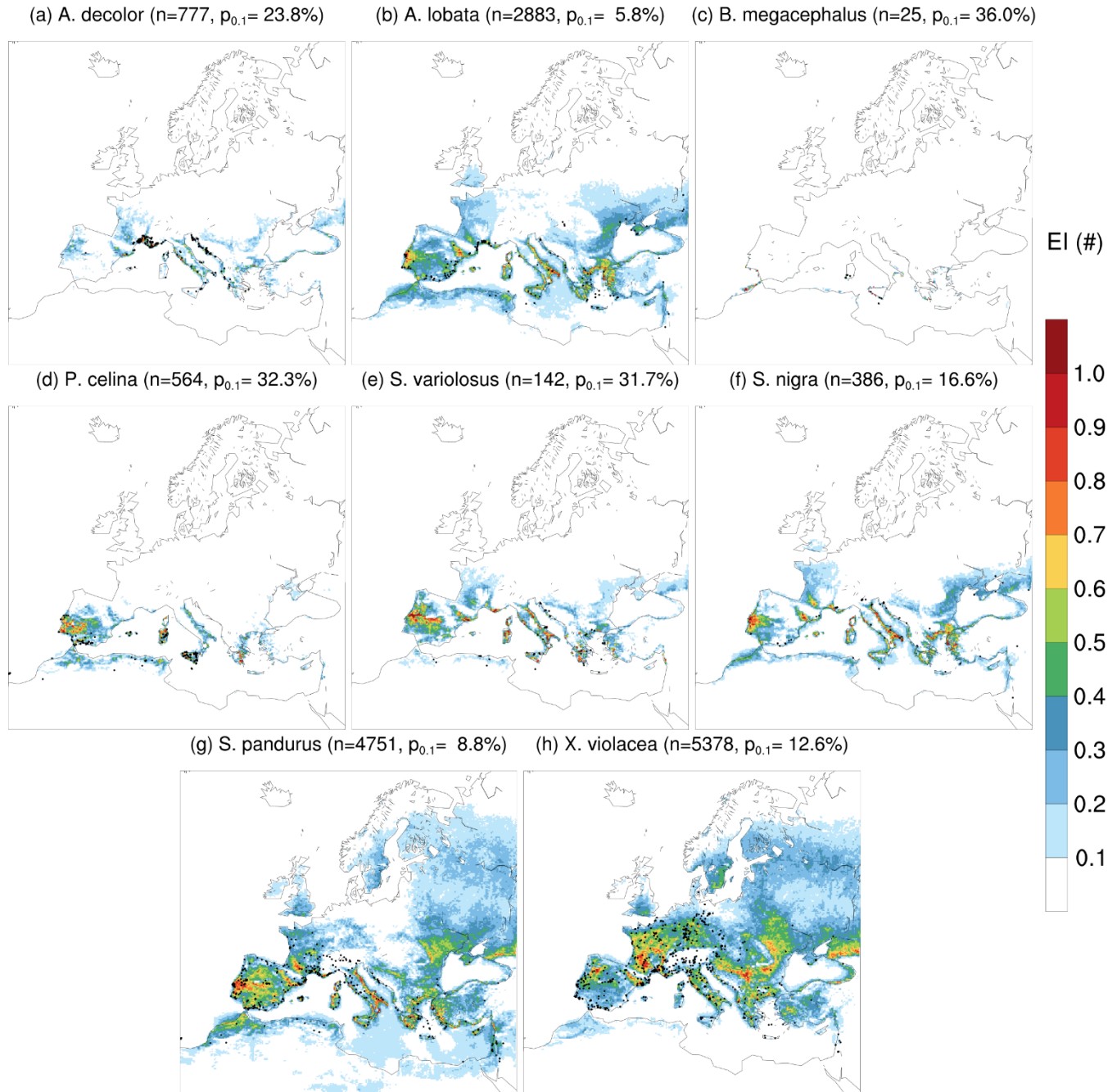

**Figure 5: The spatial distribution of $EI_s$ for all eight PSIs applied to the 1980-2010 Ens6 dataset. Individual products of each PSI are shown in Figures S9-S16.**

The conditions leading to $EI_s$ values lower than 0.1 may be a consequence of the spatial resolution of the climate data, where more complex geographic features such as streams, smaller valleys, gulleys, etc. (which can serve as micro-habitats) would not be properly represented in the dataset. The WMD03 data (Figure 6), which serves as a test for this hypothesis, also gives similar results to the previous two datasets, but with a decrease in $p_{0.1}$ for almost all species when compared to Ens6 (this is mostly consistent when analysing Ens6 within the common time period; see Figure S26). This is expressed more clearly in Figure 7, which shows the relationship of $n_s$ and $p_{0.1}$, and also highlights the differences between the three climate datasets. This reveals that the climate data, whilst resulting in some variation to the successful interpretation of the fundamental niche, is likely not a major factor in decreasing the value of $p_{0.1}$. Figure 7 clearly reveals that, instead, the most reliable results are obtained for species with a very high number of observations ($n_s$>1000). When comparing the results of each species, the reference dataset almost always produces the most reliable results. This, together with the comparable $p_{0.1}$ values of the RCM and CP dataset (which sometimes is worst for the CP data), suggests that the reducing the bias of the model data can contribute towards improving the results of this metric.

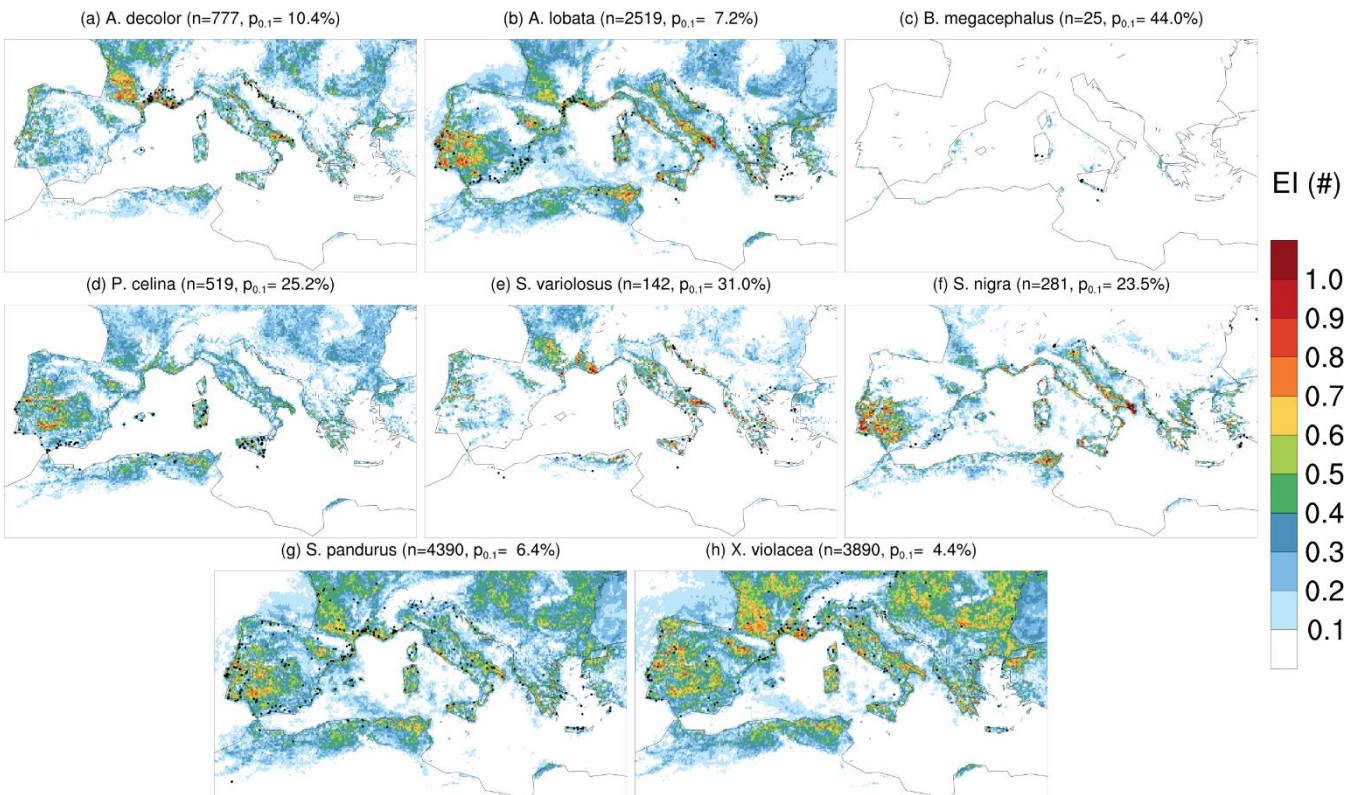

**Figure 6: The spatial distribution of $EI_s$ for all eight PSIs applied to the 1995-2004 WMD03 dataset. Individual products of each PSI are shown in Figures S17-S24.**

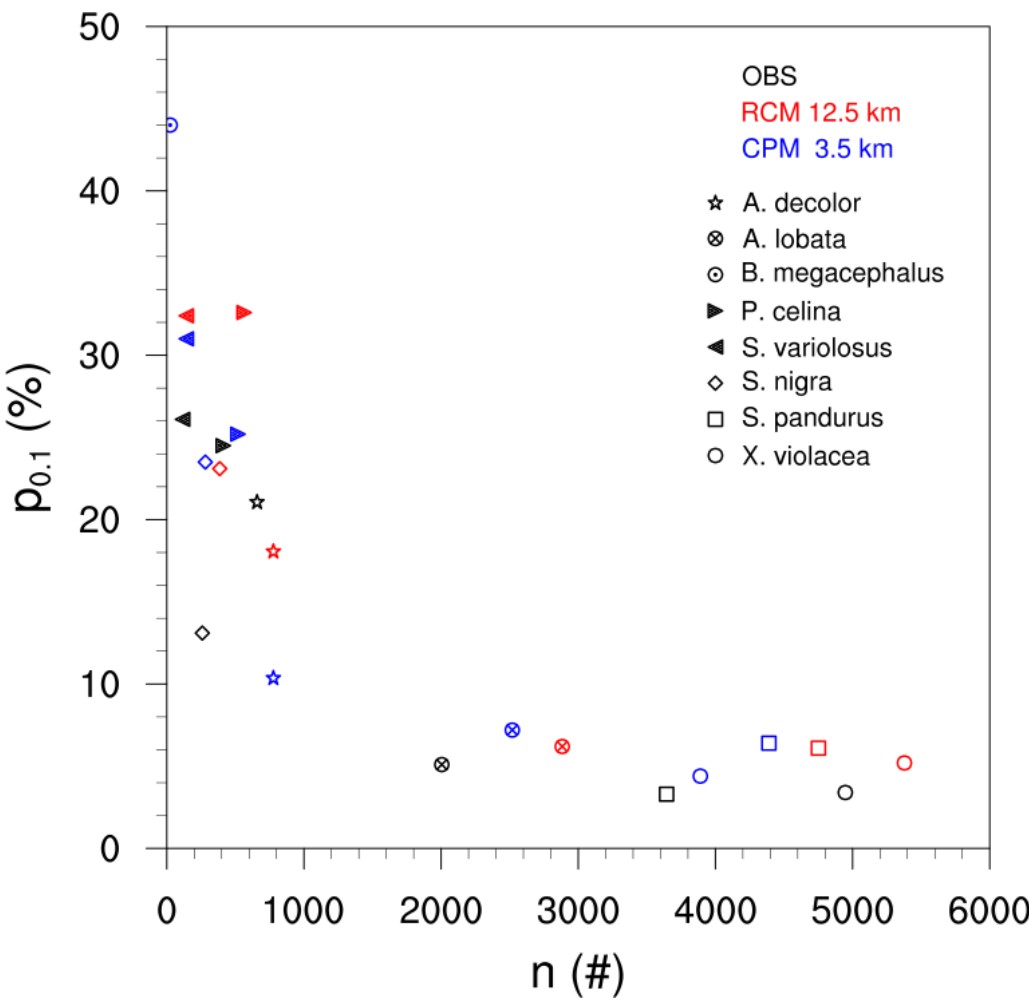

**Figure 7: The relationship between the total points $n_s$ used in each analysis and the corresponding $p_{0.1}$. Each species is presented with a unique marker, while results obtained from different climate datasets are presented with different colours.**

## 4    Conclusions

This study has introduced and applied an efficient index for assessing climate suitability for species' occurrences, with a
focus on terrestrial arthropods occurring in the Mediterranean region. Through the integration of RCM data, this research
paper outlines a methodological framework that reflects the climatological preferences of terrestrial arthropod species based
on their historically observed locations. This new metric allows a suitable representation of species distribution on the basis
of their fundamental (or climatic) niche and hence might represent an important tool to model future change in response to
climate change.

The application of a diverse range of climate data in this study has underscored the effectiveness of the proposed index in representing the fundamental niches of arthropod species across the Mediterranean region. Specifically, for species with observations exceeding 1000 points, the method captures the climatic preferences corresponding to approximately 95% of these observed points. While the index yields appreciable results with any climate dataset employed, the analysis indicates
that CP data often provides some superior outcomes compared to RCM data with a lower resolution. This distinction highlights the index's versatility and its potential for adaptation to different data sources, ensuring its applicability and usefulness in a wide range of ecological and conservation planning scenarios. The positive aspects of this research pave the way for future investigations into the impacts of climate change on biodiversity, offering a promising tool for the assessment and preservation of arthropod populations in changing environmental conditions.

Despite the promising outcomes of this study, it is important to acknowledge its limitations, particularly in the context of data availability for various arthropod species. The methodology's reliance on a significant volume of observations ($n_s > 1000$) to accurately model the fundamental niches predominantly benefits well-documented, charismatic species, such as butterflies. This criterion, unfortunately, leaves out a vast number of arthropod species that may be less well-known or
345 visually appealing but are equally or more critical from a conservation perspective. Notably, *Brachytrupes megacephalus*, falls short of the observation threshold necessary for reliable niche modelling through this index. However, this limitation also opens avenues for future research and methodological refinement. By exploring and integrating alternative data sources, there is potential to enhance the model's applicability and extend its benefits to a broader spectrum of arthropod species, ensuring that conservation efforts can be more inclusively and effectively directed.

The successful application of the proposed metric critically hinges on the selection of appropriate climate indices tailored to the specific ecological requirements of each arthropod species. Recognising the unique set of conditions that define the habitat preferences of each species necessitates an individualised approach to determining the most relevant climate indices for accurate niche modelling. During this study, to explore the metric's boundaries and potential, a uniform set of climate
indices was applied across all species examples. It is crucial to understand that the results derived from this methodology, while insightful, should not be interpreted as precise depictions of any given species's habitat. Instead, they should be viewed as illustrative examples demonstrating the metric's application. This approach underscores the necessity for nuanced, species-specific research to fully leverage the metric's capabilities in accurately representing the ecological niches of arthropods, thereby reinforcing the importance of customisation in the pursuit of ecological understanding and conservation
efforts.

In conclusion, the metric introduced in this study holds the potential for application across a variety of climate scenarios, including future projections from the CORDEX ensembles. Such applications promise to yield valuable insights into the

direct impacts of climate change on the ecological niches of species at risk. Envisioned as the basis for follow-up studies, this metric could significantly enhance our comprehension of how climate variability affects biodiversity and ecosystem dynamics. By delineating potential shifts in the fundamental niches of key ecological actors, this research not only advances our understanding of the intricate relationships within ecosystems under the pressure of climate change but also provides practical guidance for conservation strategies. These strategies aim to address and mitigate the negative consequences of environmental changes, thereby supporting the resilience of biodiversity in the face of impending climatic challenges.

## Funding

This work was supported by the Marie Skłodowska-Curie Actions Grant Agreement 101062427.

## Acknowledgements

The authors would like to express their gratitude to Arthur Lamoliere, Simone Cutajar, and Cyril Caminade for their insightful and constructive discussions that greatly enhanced this work. The authors acknowledge the E-OBS dataset from the EU-FP6 project UERRA (http://www.uerra.eu) and the Copernicus Climate Change Service, and the data providers in the ECA&D project (https://www.ecad.eu). They also acknowledge the World Climate Research Programme's Working Group on Regional Climate, and the Working Group on Coupled Modelling, former coordinating body of CORDEX and responsible panel for CMIP5. They also thank the climate modelling groups (listed in Table 3 of this paper) for producing and making available their model output, as well as the Earth System Grid Federation infrastructure, an international effort led by the U.S. Department of Energy's Program for Climate Model Diagnosis and Intercomparison, the European Network for Earth System Modelling and other partners in the Global Organisation for Earth System Science Portals (GO-ESSP). ChatGPT (GPT-3.5, OpenAI's large-scale language-generation model) has been used to improve the writing style of limited parts of this Article. JMC reviewed, edited, and revised the ChatGPT generated texts and takes ultimate responsibility for the content of this publication. The authors sincerely thank the reviewers for their constructive suggestions, which greatly contributed to improving the quality of this study.

## Data Access

The scripts used for the analysis in this study are available on GitHub at: https://github.com/ciarloj/PALEOSIM. The data from the CP simulations is freely available on Zenodo: https://doi.org/10.5281/zenodo.14537964.

## Competing Interests

The authors declare that they have no conflict of interest.

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
