# Peer review of "A climate suitability index for species distribution modelling applied to terrestrial arthropods in the Mediterranean Region"

_EGUsphere, 2024_

## Referee Comment (RC1)

**Review of the article:**

A climate suitability index for ecological habitats applied to terrestrial arthropods in the Mediterranean Region

**General Comments:**

This study presents a simple index to assess the climatic suitability of terrestrial arthropod populations in the Mediterranean and aims to fill the gap in our understanding of how climate change could affect their fundamental niches. The results highlight that the index is most reliable for species with significant volume of observations (ns >1000), and high resolution climate datasets. The authors claim that the proposed index is a proof of concept for its potential utility in guiding conservation strategies and mitigating the adverse effects of climate change on arthropod habitats.

The paper is consistent with the aims and scope of the journal. The manuscript is well written; clear, and easy to understand and addresses an interesting and not extensively explored topic. I have, however, some concerns with regard to the specific aims of the article and the methodological framework in use, which will be discussed below.

**Specific Comments:**

This study uses the index to assess the climatic suitability of terrestrial arthropod populations in the Mediterranean in the recent past (1980-2010 and 1995-2004) and does not provide any information about the future. Therefore, the following sentences are misleading and should be amended to accurately reflect the study.

line 19: *"aiming to shed light on how climate change could affect their fundamental niches."*
line 69: *"evaluate the effects of climate change on terrestrial arthropod habitats,.."*
line 77: *"a critical analysis given the anticipated direct impacts of climate change on countless species."*

Authors should emphasize that the study evaluates the performance of the metric in the recent past using hindcast RCM data, and if it is found to perform well, it could be used to assess the impact of climate change on terrestrial arthropod habitats. The time periods (1980-2010 and 1995-2004) should be clearly stated in the Data & Methods section.

Another concern is the temporal and spatial resolution. Authors should provide datasets with the same temporal and spatial resolution to be compared. This means that the Ens6 and EOBs datasets should also be examined in the period 1995-2004 in order to be compared with the WMD03 dataset. In the Data & Methods section, authors should explain how they get climate datasets to the same spatial resolution so that they can be compared and how the Ens6 has been calculated?

The article does not discuss the ideal geographical-climatic conditions for terrestrial arthropods in the Mediterranean in the present and how they might expand in the future. Where do we see most populations and where are their numbers expected to increase due to climate change?

What were the criteria for selecting the climate indices used in the study? Have you also checked if RCMs perform well in the representation of the selected indices? I would suggest a state-of-the-art literature review on this topic in the introduction section.

How do you explain that p0.1 increases for n>4000 (see Figure 7)?

How do you explain that in Fig.3 a-c the EIs product is almost everywhere close to 1?

**Technical corrections**

**1 Introduction**

**1: line 30: Please add references that relate to the Mediterranean basin.**

**2: line 38: I suggest you to move this citation list (Doblas-Reyes et al., 2021; Gutiérrez et al., 2021; Ranasinghe et al., 2021) at the end of the sentence.**

**3: line 60: Please change the *"Coppola, Nogherotto, et al., 2021"* to *"Coppola et al., 2021"* throughout the text**

**4: line 63: Please add references that relate to the Mediterranean basin.**

**5: line 67: Please add references.**

**6: line 71: what does this mean "*or experiments*"?**

**2 Data & Methods**

**7: line 118: $n_s$ should be consistent throughout the text**

**8: line 144: *please change "by the ECMWF-ERAINT" to "by the European Centre for Medium-Range Weather Forecasts ERA-Interim reanalysis (ECMWF-ERAINT)"**

**9: line 154: please change *"using the RegCM5"* to *"using the fifth generation regional climate modeling system, RegCM5"**

**10: line 155: please change *"driven by the ECMWF-ERA5 reanalysis"* to *"driven by the fifth generation ECMWF reanalysis for the global climate and weather, ERA5"**

**11: line 181: please indicate in the methods section that the EOBs is considered as the reference dataset**

**12: line 181: Please explain how the data sets were combined.**

**13: line 204: Please change *"this describes"* to *"they represent"**

**14: line 244: *"..but is almost consistently better than the Ens6." How is it better than Ens6?* Can you elaborate on that?**

**Figures**

The resolution of the figures should be increased and the titles should be larger. Please add latitude and longitude to the maps.

---

## Author Comment (AC1)

We are grateful for the reviewer's constructive comments. Please find our responses below (in red text).

**Specific Comments:**

This study uses the index to assess the climatic suitability of terrestrial arthropod populations in the Mediterranean in the recent past (1980-2010 and 1995-2004) and does not provide any information about the future. Therefore, the following sentences are misleading and should be amended to accurately reflect the study.

line 19: *"aiming to shed light on how climate change could affect their fundamental niches."*

line 69: *"evaluate the effects of climate change on terrestrial arthropod habitats,.."*

line 77: *"a critical analysis given the anticipated direct impacts of climate change on countless species."*

Authors should emphasize that the study evaluates the performance of the metric in the recent past using hindcast RCM data, and if it is found to perform well, it could be used to assess the impact of climate change on terrestrial arthropod habitats. The time periods (1980-2010 and 1995-2004) should be clearly stated in the Data & Methods section.

We agree with the suggested clarifications and will implement them in the next version of the manuscript.

Another concern is the temporal and spatial resolution. Authors should provide datasets with the same temporal and spatial resolution to be compared. This means that the Ens6 and EOBs datasets should also be examined in the period 1995-2004 in order to be compared with the WMD03 dataset.

We also agree with the reviewer that the temporal resolution should match for all datasets when comparing the different datasets. This is especially important for Figure 7 – and we can use the 1995-2004 time period for the EOBS and Ens6 datasets to match the time period of WMD03. However, we would suggest keeping the complete 1980-2010 time periods for the remainder of the analysis (Figures 1, 3-5) as a 30 year time period is more reliable climatologically than a 10 year period (unfortunately extending beyond 10 years for the WMD03 dataset is not attainable at this time).

In the Data & Methods section, authors should explain how they get climate datasets to the same spatial resolution so that they can be compared and how the Ens6 has been calculated?

We agree with the reviewer that a clarification of this material is required. In the next manuscript we will clarify as follows (within Section 2.2): All ensemble members are at the same spatial resolution, however 2 members (CNRM-ALADIN63, and ICTP-RegCM4-6) required interpolation from their native grid to a common grid using a nearest-neighbour approach. To minimize errors, the indices listed in Table 2 were calculated individually before any interpolation. Furthermore, the final Ens6 product was obtained with the use of an ensemble mean of the indices associated with each member.

The article does not discuss the ideal geographical-climatic conditions for terrestrial arthropods in the Mediterranean in the present and how they might expand in the future. Where do we see most populations and where are their numbers expected to increase due to climate change?

Determining the ideal conditions of each variable (in their respective units, not relative EI values) associated with a particular species is possible, and would be informative to users. We would add necessary change to the scripts to provide this information for users. It is worth exploring this within the study as a confirmation (to users and readers) that no abnormal conditions are being defined as

"ideal". Such values could be included in table within the supplementary information, however, we would avoid depending on these quantities within the main text so as not to mislead the readers, given that we are using the same variables for each species.

We would also add some comments within the main text, based on the results in Figure 3, how the distributions in Figures 3a-h can be used to extrapolate on expected changes due to climate change. For instance, with the expected aridification of the Mediterranean, one would expect that the impacted regions would become less hospitable to the species in question given the importance of prsum (Figure 3b). We would also comment on tasmean (Figure 3e), where we would expect the ideal conditions to migrate north, as southern regions become less hospitable with increasing temperatures, and northern regions become more hospitable. Speculating on the remaining indices would more specialized assessment based on future simulations, but the combination of prsum and tasmean alone could already represent a potentially dangerous reduction in habitable zones.

What were the criteria for selecting the climate indices used in the study?

In section 2.2 we would add this information. The first variables considered to assess the environmental conditions preferred by a given species were temperature (due to its importance to an organism's metabolism) and precipitation (due to the importance of a water source). Given the importance of these variables and their variability throughout the year, the mean conditions together with upper and lower extreme conditions were also deemed important. Thus, indices were selected that represent these conditions for both temperature and precipitation. Given the size of arthropods, average windspeed was also included. Finally, as organisms are known to have a preference to specific altitudes, elevation was also included. Beyond the proof-of-concept these criteria can be used to list starting indices but should not be used as strict rules to be satisfied.

Have you also checked if RCMs perform well in the representation of the selected indices? I would suggest a state-of-the-art literature review on this topic in the introduction section.

The evaluation of the RCMs was limited to the Ensemble means given the extensive research on individual EURO-CORDEX members conducted in Teichmann et al. (2021), Coppola, et al. (2021), and other studies. However, we will include the expand appropriately on this matter within the literature review.

How do you explain that p0.1 increases for n>4000 (see Figure 7)?

The higher values are associated with the RCM results for S. pandurus and X. violacea, which suggests that this "behaviour" may be attributed to the dataset, resulting in significant variation within the p0.1 value. It is possible that the bias in rx1day and/or hwfi (Figure 1), although small may result in variation within the p0.1 value – this could be explored by applying bias correction to the End6 dataset and evaluating the change. The change may also be related to the way subgroups of n "cluster" within each grid cell – hence given the lower resolution of RCM with respect to the CPM, or different grid of RCM with respect to OBS, this could result in variation p0.1. Verifying this hypothesis is more challenging, but we could gather some insight by analysing a subsample within a smaller area.

How do you explain that in Fig.3 a-c the EIs product is almost everywhere close to 1?

Given that the lower limit of the index of consecutive dry days is 0, these minimal drought conditions likely do not pose any additional stress to the species in question. We would be very happy to include this clarification, as further interpretation of the Figure.

**Technical corrections**

All technical corrections are achievable and will be addressed within the next manuscript.

**1 Introduction**

**1: line 30: Please add references that relate to the Mediterranean basin.**

**2: line 38: I suggest you to move this citation list (Doblas-Reyes et al., 2021; Gutiérrez et al., 2021; Ranasinghe et al., 2021) at the end of the sentence.**

**3: line 60: Please change the *"Coppola, Nogherotto, et al., 2021"* to *"Coppola et al., 2021"* throughout the text**

**4: line 63: Please add references that relate to the Mediterranean basin.**

**5: line 67: Please add references.**

**6: line 71: what does this mean "*or experiments*"?**

**2 Data & Methods**

**7: line 118: *ns* should be consistent throughout the text**

**8: line 144: *please change "by the ECMWF-ERAINT" to "by the European Centre for Medium-Range Weather Forecasts ERA-Interim reanalysis (ECMWF-ERAINT)"**

**9: line 154: please change *"using the RegCM5"* to *"using the fifth generation regional climate modeling system, RegCM5"**

**10: line 155: please change *"driven by the ECMWF-ERA5 reanalysis"* to *"driven by the fifth generation ECMWF reanalysis for the global climate and weather, ERA5"**

**11: line 181: please indicate in the methods section that the EOBs is considered as the reference dataset**

**12: line 181: Please explain how the data sets were combined.**

**13: line 204: Please change *"this describes"* to *"they represent"**

**14: line 244: "*..but is almost consistently better than the Ens6." How is it better than Ens6?* Can you elaborate on that?**

**Figures**

The resolution of the figures should be increased and the titles should be larger. Please add latitude and longitude to the maps.

---

## Author Response (AR1)

We are grateful for the constructive comments of both reviewers, they have been instrumental in improving the quality of this study. We would also like to acknowledge that thanks to the reviewer's comments we noticed a mistake in the percentage bias scripts, which impacted the results shown. This has been corrected and explained within the manuscript. Please find our responses below (in red text).

**SD Comments:**

This study uses the index to assess the climatic suitability of terrestrial arthropod populations in the Mediterranean in the recent past (1980-2010 and 1995-2004) and does not provide any information about the future. Therefore, the following sentences are misleading and should be amended to accurately reflect the study.

line 19: *"aiming to shed light on how climate change could affect their fundamental niches."*

line 69: *"evaluate the effects of climate change on terrestrial arthropod habitats,.."*

line 77: *"a critical analysis given the anticipated direct impacts of climate change on countless species."*

Authors should emphasize that the study evaluates the performance of the metric in the recent past using hindcast RCM data, and if it is found to perform well, it could be used to assess the impact of climate change on terrestrial arthropod habitats. The time periods (1980-2010 and 1995-2004) should be clearly stated in the Data & Methods section.

We agree with the suggested clarifications and have implemented them in the manuscript.

Another concern is the temporal and spatial resolution. Authors should provide datasets with the same temporal and spatial resolution to be compared. This means that the Ens6 and EOBs datasets should also be examined in the period 1995-2004 in order to be compared with the WMD03 dataset.

We also agree with the reviewer that the temporal resolution should match for all datasets when comparing the different datasets. We have kept the complete 1980-2010 time periods for the first part of the analysis (Figures 1, 3-5) as a 30 year time period is more reliable climatologically than a 10 year period (unfortunately extending beyond 10 years for the WMD03 dataset is not attainable at this time). But we prepared the 10-year version of Figures 4 and 5 in the supplementary material, and applied this to Figure 7.

In the Data & Methods section, authors should explain how they get climate datasets to the same spatial resolution so that they can be compared and how the Ens6 has been calculated?

We agree with the reviewer that a clarification of this material is required. We have clarified as follows: All ensemble members are at the same spatial resolution, however 2 members (CNRM-ALADIN63, and ICTP-RegCM4-6) required interpolation from their native grid to a common grid using a nearest-neighbour approach. To minimize errors, the indices listed in Table 2 were calculated individually before any interpolation. Furthermore, the final Ens6 product was obtained with the use of an ensemble mean of the indices associated with each member.

The article does not discuss the ideal geographical-climatic conditions for terrestrial arthropods in the Mediterranean in the present and how they might expand in the future. Where do we see most populations and where are their numbers expected to increase due to climate change?

We have prepared the actual values of the statistical descriptors in tables and added them to the supplementary material. We have commented on these in the manuscript, that their interpretation within the study should be used as a confirmation that no abnormal conditions are being defined as "ideal".

We have also add some comments within the main text, based on the results in Figure 3, how the distributions in Figures 3a-h can be used to extrapolate on expected changes due to climate change.

What were the criteria for selecting the climate indices used in the study?

In section 2.2 we would add this information. "The first variables considered to assess the environmental conditions preferred by a given species were temperature (due to its importance to an organism's metabolism) and precipitation (due to the importance of a water source). Given the importance of these variables and their variability throughout the year, the mean conditions together with upper and lower extreme conditions were also deemed important. Thus, indices were selected that represent these conditions for both temperature and precipitation. Given the size of arthropods, average windspeed was also included. Finally, as organisms are known to have a preference to specific altitudes, elevation was also included. Beyond the proof-of-concept these criteria can be used to list starting indices but should not be used as strict rules to be satisfied."

Have you also checked if RCMs perform well in the representation of the selected indices? I would suggest a state-of-the-art literature review on this topic in the introduction section.

We expanded in various parts of the manuscript with appropriate comments and citations on the ensemble, and added the bias of the individual members in the supplementary material.

How do you explain that p0.1 increases for n>4000 (see Figure 7)?

Following the implementation of the 10-year EOBS and Ens6 analysis to this image, this increase was no longer visible.

How do you explain that in Fig.3 a-c the EIs product is almost everywhere close to 1?

Given that the lower limit of the index of consecutive dry days is 0, these minimal drought conditions likely do not pose any additional stress to the species in question. We have included this clarification in the manuscript.

**Technical corrections**

All technical corrections have been addressed within the manuscript.

**1 Introduction**

**1: line 30: Please add references that relate to the Mediterranean basin.**

**2: line 38: I suggest you to move this citation list (Doblas-Reyes et al., 2021; Gutiérrez et al., 2021; Ranasinghe et al., 2021) at the end of the sentence.**

**3: line 60: Please change the *"Coppola, Nogherotto, et al., 2021"* to *"Coppola et al., 2021"* throughout the text**

**4: line 63: Please add references that relate to the Mediterranean basin.**

**5: line 67: Please add references.**

**6: line 71: what does this mean "*or experiments*"?**

**2 Data & Methods**

**7: line 118: *ns* should be consistent throughout the text**

**8: line 144: *please change "by the ECMWF-ERAINT" to "by the European Centre for Medium-Range Weather Forecasts ERA-Interim reanalysis (ECMWF-ERAINT)"**

**9: line 154: please change *"using the RegCM5"* to *"using the fifth generation regional climate modeling system, RegCM5"**

**10: line 155: please change *"driven by the ECMWF-ERA5 reanalysis"* to *"driven by the fifth generation ECMWF reanalysis for the global climate and weather, ERA5"**

**11: line 181: please indicate in the methods section that the EOBs is considered as the reference dataset**

**12: line 181: Please explain how the data sets were combined.**

**13: line 204: Please change *"this describes"* to *"they represent"**

**14: line 244: "*..but is almost consistently better than the Ens6.*" How is it better than Ens6? Can you elaborate on that?**

**Figures**

The resolution of the figures should be increased and the titles should be larger. Please add latitude and longitude to the maps.

The sizes and resolutions of the Figures were increased, however, the latitude and longitude were not added. While modifying the images, the resultant images were restricted to account for latitude and longitude which reduced the visible size of the images. Since the geography of the region is very understandable, and in order to maximize the visible space for the actual images, the latitude and longitude were omitted.

**AR Comments:**

This research presents a novel approach to habitat suitability modelling based on climatic parameters. The approach is based on the calculation of an index which is conceptually simple, theoretically well-founded, easy to understand and versatile.

The authors clearly explain the advantages and the limits of their approach. The new approach is described in a clear and detailed way, and then applied to arthropod data to show how it works.

The study is surely interesting and has a great potential, especially given the importance of modelling species' climatic niches to forecast how species distributions will be affected by the ongoing climate change. Although the authors developed their approach for modelling arthropod distributions, it can be applied to a diversity of taxa, and hence has a general validity.

The results of the applications shown in the manuscript are clear and convincing.

The manuscript is well-organised and easy to follow.

In general, I do not have any major concern, but I can offer a few suggestions which might improve manuscript readability and appeal.

SPECIFIC COMMENTS

Title

I think the title does not adequately explain the importance of the paper. Also, the expression "for ecological habitats" sounds me not very appropriate. I would suggest something like: "A new climate suitability index for species distribution modelling applied to terrestrial arthropods in the Mediterranean Region".

The title suggestion is received with gratitude and was applied in this version.

Introduction

The Introduction is very well articulated. However, at lines 24-28, I suggest introducing some references supporting these sentences. In fact, they may appear obvious to any reader sufficiently familiar with arthropods, but not for people working on other subject. Thus, including a few, very basic references, may be useful for a broader readership.

69: I'm not convinced that the sentence "This study utilises RCM data to evaluate the effects of climate change on terrestrial arthropod habitats, introducing a novel" really describes the contents of this manuscript, as the authors do not use their approach to model future distributions, but current ones (although it can be actually used also for modelling future distributions). Thus, I would suggest writing: "This study utilises RCM data to evaluate the influence of climate parameters on terrestrial arthropod distribution patterns, introducing a novel".

74: Similarly, I suggest changing "and demonstrate its applicability in assessing climate change impacts onarthropod habitats." to "and demonstrate its applicability in modelling arthropod species distribution on the basis of their climatic niches."

The suggestions above are understandable and have been applied in this version.

Methos

Methos are clearly illustrated. However, I think that the paper might more appealing, especially for a broader readership, if the use of the selected species is better justified. Why did you used these arthropods, and not others? For example, I would explain that these are representative of different sampling densities, trophic roles, dispersal capabilities, climatic preferences, and so on. This information might be presented in the main text and neatly summarised in Table 1 (where differences in sampling density is already apparent).

Also, it is not well explained what the percentage bias is. Please, introduce a clear definition.

The suggestions above are understandable and have been applied in this version.

Typos/style/ miscellanea

Although is accepted to consider data as singular, it is actually the plural of datum. Thus, I suggest using it as plural. So, at line 52: "resolution; Karger et al., 2017) is preferred" -> "resolution; Karger et al., 2017) are preferred"

While we acknowledge that data is etymologically the plural of datum, it is also widely used as a collective noun (such as council, team, public, or swarm), as noted by Oxford Dictionaries. In such contexts, data is often treated as a singular, though its usage has sparked ongoing discussion in academic circles. With respect for these differing perspectives, we will continue to use data as a collective noun in this context.

I'm aware that most people use species' also to indicate possession when the noun species is treated as singular. However, in such a case, species's should be used (see Chicago Manual of Style):

Species': Used to indicate possession when the noun species is treated as plural. Example: "The species' habitats were destroyed." (The habitats belonging to multiple species were destroyed.)

Species's: Used to indicate possession when the noun species is treated as singular. Example: "The species's survival is at risk." (The survival of a single species is at risk.)

Thus, line 99 climatological component of a species' ecological niche-> climatological component of a species's ecological niche

The suggestions above are understandable and have been applied in this version.

Fundamental niche and realised niche are well known terms in ecology, but to help readers not familiar with their meaning I suggest providing a definition. In particular, I suggesting clearly indicating that your use of fundamental niche refer to the climatic niche.

15 This study introduces a simple index designed to assess the climate suitability of ecological habitats, with a specific focus on terrestrial Mediterranean arthropods -> This study introduces a simple index designed to model species' distribution on the basis of their climatic nice, with a specific focus on terrestrial Mediterranean arthropods

62: Delete "small and". I would not define the Mediterranean a small area.

65-66: I suggest including a few references supporting the association of arthropods with microclimates

75: "on certain terrestrial arthropod habitats, a critical analysis" -> ""on the distribution of certain terrestrial arthropod, a critical analysis"

80: I think that "necessary" is not the best choice here. I would say: "Hence, a collection of locations where the organism was observed can describe the range of climate parameters of its fundamental niche"

84: indices include, annual-> indices include annual

87: The ideal conditions for s would occur when -> The most appropriate conditions for s would occur when

89: the climate index becomes less ideal, -> the climate index identifies less favourable conditions,

The suggestions above are understandable and have been applied in this version.

109: I would avoid using "to test", as this is not a statistical test.

We have changed "test" to "evaluate" in this version.

117: add reference: Buzzetti, F.M., Fontana, P., Hochkirch, A., Kleukers, R., Massa, B. & Odé, B. 2016. Brachytrupes megacephalus (Europe assessment). The IUCN Red List of Threatened Species 2016: e.T64550733A70738413. https://www.iucnredlist.org/species/64550733/70738413 Accessed on 10 January 2025.

The suggestions above are understandable and have been applied in this version.

118 is ns = starting sample size (i.e., number of occurrences)?

Yes, this was clarified in this version.

128: impacts -> influence

129: indices (Coppola, Nogherotto, et al., 2021 -> indices (Coppola et al., 2021

The suggestions above are understandable and have been applied in this version.

157-158: This part is not very clear to me.

This is a mistake in the referencing system. It have been applied in this version.

221: Table 1, provide the opportunity -> Table 1 provide the opportunity

248: Figure 7 clearly reveals that instead, the -> Figure 7 clearly reveals that, instead, the

259: assessing the climate suitability of ecological habitats, with a -> assessing climate suitability for species' occurrences

263: This sentence is not very appropriate, as you did not use the new approach to predict future changes. I would reformulate it to stress that the new approach allows a suitable representation of species distribution on the basis of their climatic niche and hence might represent an important tool to model future change in response to climate change.

386: Recognizing -> Recognising (British English)

290: of any given species' habitat -> of any given species's niche

293: customization -> customisation (British English)

The suggestions above are understandable and have been applied in this version.

---

## Referee Report (RR1)

**Review of the article:**

egusphere-2024-1954

A climate suitability index for species distribution modelling applied to terrestrial arthropods in the Mediterranean Region

James M. Ciarlo`, Monique Borg Inguanez, Erika Coppola, Aaron Micallef, and David Mifsud

The authors have adequately addressed the comments made during the first round of peer review, and the manuscript has been substantially improved, so I would recommend that it be accepted after some minor corrections, mostly related to the ensemble mean and the use of the EURO-CORDEX hindcast simulations, which I think is important to clarify for other studies.

Minor comments:

The author claimed that the ensemble was constructed only from simulations that provided the parameters necessary to construct the indices described in Table 2. However, to my knowledge, the EURO-CORDEX data are consistent in the variables they provide and their spatial resolution (0.11°), which means that all simulations participating in the hindcast experiment provide all the variables listed here: http://is-enes-data.github.io/CORDEX_variables_requirement_table.pdf from which the authors could calculate the indices. Also with a quick search on the Earth System Grid Federation https://esgf-metagrid.cloud.dkrz.de/search I could not find the RegCM4-6 data. I think it's very important to clarify what criterion was used for a model to be included as a member in the ensemble. Finally, I don't understand why there were two models, the CNRM-ALADIN63 and the ICTP-RegCM4-6, that needed interpolation, since all EURO-CORDEX hindcast data are provided at the same spatial resolution as mentioned above. It seems that the later models are not driven by ERA-interim, but by general circulation models, and if this is the case, then the ensemble model should be rebuilt using only hindcast simulations.

**1: line 159: "The observations are the 30-year (1980-2010)" Please change to 31-year everywhere in the text.**

---

## Author Response (AR2)

We are grateful for your thoughtful and constructive comments, and we hope that the responses provided below address the remaining queries with sufficient clarity.

**SD Comments v2:**

The authors have adequately addressed the comments made during the first round of peer review, and the manuscript has been substantially improved, so I would recommend that it be accepted after some minor corrections, mostly related to the ensemble mean and the use of the EURO-CORDEX hindcast simulations, which I think is important to clarify for other studies.

Minor comments:

The author claimed that the ensemble was constructed only from simulations that provided the parameters necessary to construct the indices described in Table 2. However, to my knowledge, the EURO-CORDEX data are consistent in the variables they provide and their spatial resolution (0.11°), which means that all simulations participating in the hindcast experiment provide all the variables listed here: http://is-enes-data.github.io/CORDEX_variables_requirement_table.pdf from which the authors could calculate the indices.

We appreciate the reviewer's observation regarding the expected consistency of EURO-CORDEX simulations in terms of variable availability and resolution, in line with established CORDEX protocols. However, despite these formal requirements, there were practical limitations at the time we initiated the analysis. Specifically, several ESGF nodes were offline, making it impossible to access some simulations (for instance, the current issue with RegCM4-6 outputs, as discussed below).

Given these accessibility constraints, we proceeded with the subset of simulations that were readily available at that time. As the issue persisted for a while and considering that the current study is intended as a proof-of-concept, the primary objective was to demonstrate the effectiveness of the methodology rather than to exhaustively include all possible model runs. Expanding the ensemble would likely not have significantly altered the results, nor would it have influenced the conclusions of the study.

For these reasons, we respectfully maintain that it is not necessary to expand the ensemble further at this stage.

Also with a quick search on the Earth System Grid Federation https://esgf-metagrid.cloud.dkrz.de/search I could not find the RegCM4-6 data.

The RegCM4-6 data is listed via the ESGF LiU server hosted at https://esg-dn1.nsc.liu.se/search/cordex/, although it may currently be experiencing accessibility issues. It is likely that the RegCM4-6 data is temporarily unavailable due to disruptions at the ESGF-ICTP node (esgf-ictp.hpc.cineca.it), which was hosted on the Marconi system at the CINECA supercomputing centre. This system sustained significant damage during the severe flooding in Bologna on October 20th, 2024, resulting in prolonged outages for both computing and data access services. We were informed that this is a temporary issue and expect data access to resume but have no information regarding the timeline.

I think it's very important to clarify what criterion was used for a model to be included as a member in the ensemble.

In light of the comments above, we have clarified the sentence in the manuscript describing the ensemble selection by explicitly stating that the included models were also those "available through ESGF nodes at the time data collection began."

Finally, I don't understand why there were two models, the CNRM-ALADIN63 and the ICTP-RegCM4-6, that needed interpolation, since all EUROCORDEX hindcast data are provided at the same spatial resolution as mentioned above.

We acknowledge the reviewer's observation and agree that all EURO-CORDEX simulations conform to a nominal resolution of 0.11°. However, modelling institutions retain flexibility in choosing the projection and some domain configuration best suited to their models. As a result, while all simulations cover the common EURO-CORDEX domain, some differences (e.g., projection and larger grid sizes) may be sometimes present. In these cases, interpolation was required to ensure consistency across grids for subsequent analysis. To minimize errors, the indices listed in Table 2 were calculated individually before any interpolation, as stated in the manuscript.

It seems that the later models are not driven by ERA-interim, but by general circulation models, and if this is the case, then the ensemble model should be rebuilt using only hindcast simulations.

Indeed, there is an important distinction between simulations driven by reanalysis datasets such as ERA-Interim (referred to as evaluation runs) and those driven by general circulation models (GCMs), which include historical and future climate projections.

For the purposes of this study, we specifically selected ERA-Interim-driven simulations to ensure consistency across the different data-sets and to maintain alignment with observation data. Reanalysis-driven simulations are generally preferred for evaluation studies, as they are constrained by observations and therefore offer a more realistic representation of the historical climate compared to GCM-driven simulations, which reflect internally consistent but divergent "mirror world" realizations.

Given the aims of this work and the nature of the analysis, rebuilding the ensemble to include only GCM-driven simulations would not be appropriate and would reduce the comparability and reliability of the results in relation to observed climate variability.

**1: line 159: "The observations are the 30-year (1980-2010)" Please change to 31-year everywhere in the text.**

Changes made as suggested.